# Effect of Dietary Phenolic Compounds on Incidence of Cardiovascular Disease in the SUN Project; 10 Years of Follow-Up

**DOI:** 10.3390/antiox11040783

**Published:** 2022-04-14

**Authors:** Zenaida Vázquez-Ruiz, Estefanía Toledo, Facundo Vitelli-Storelli, Leticia Goni, Víctor de la O, Maira Bes-Rastrollo, Miguel Ángel Martínez-González

**Affiliations:** 1Department of Preventive Medicine and Public Health, Instituto de Investigación Sanitaria de Navarra (IdiSNA), University of Navarra, 31008 Pamplona, Spain; zvazquez@unav.es (Z.V.-R.); etoledo@unav.es (E.T.); lgoni@unav.es (L.G.); vdelao@unav.es (V.d.l.O.); mbes@unav.es (M.B.-R.); 2Biomedical Research Network Centre for Pathophysiology of Obesity and Nutrition (CIBEROBN), Carlos III Health Institute, 28029 Madrid, Spain; 3Grupo de Investigación en Interacciones Gen-Ambiente y Salud (GIIGAS), Instituto de Biomedicina (IBIOMED), Universidad de León, 24071 León, Spain; fvits@unileon.es; 4Department of Nutrition, Harvard T. H. Chan School of Public Health, Harvard University, Boston, MA 02115, USA

**Keywords:** (poly)phenols, flavonoids, cardiovascular disease, cardioprotective, SUN cohort

## Abstract

The health benefits of plant-based diets have been reported. Plant-based diets found in Spain and other Mediterranean countries differ from typical diets in other countries. In the Mediterranean diet, a high intake of phenolic compounds through olives, olive oil, and red wine may play an important role in cardiovascular prevention. Prospective studies carried out in Mediterranean countries may provide interesting insights. A relatively young Mediterranean cohort of 16,147 Spanish participants free of cardiovascular disease (CVD) was followed (61% women, mean (SD) age 37(12) years at baseline) for a median of 12.2 years. Dietary intake was repeatedly assessed using a 136-item validated food frequency questionnaire, and (poly)phenol intake was obtained using the Phenol-Explorer database. Participants were classified as incident cases of CVD if a medical diagnosis of myocardial infarction, stroke, or cardiovascular death was medically confirmed. Time-dependent Cox regression models were used to assess the relationship between (poly)phenol intake and the incidence of major CVD. A suboptimal intake of phenolic compounds was independently associated with a higher risk of CVD, multivariable-adjusted hazard ratio for the lowest versus top 4 quintiles: 1.85 (95% CI: 1.09–3.16). A moderate-to-high dietary intake of phenolic compounds, especially flavonoids, is likely to reduce CVD incidence in the context of a Mediterranean dietary pattern.

## 1. Introduction

Cardiovascular disease (CVD) is the largest cause of mortality in Western countries, and 17.9 million people globally died in 2019 from CVDs (32% of all-cause global mortality), mostly from coronary heart disease and stroke [1]. According to the World Health Organization, most CVDs can be avoided by addressing risk factors such as obesity, hypertension, dyslipidemia, diabetes, tobacco use, sedentary lifestyle, alcohol abuse, and high consumption of ultra-processed foods [2,3]. Nutritional compounds present in foods have a key influence on human health, even though the relationship between different dietary compounds and health is complex and intertwined with each other, as well as with other health determinants. Available evidence supports that some dietary strategies, such as the traditional Mediterranean dietary pattern, rich in extra-virgin olive oil, fruit, and vegetables may delay or prevent chronic illness, such as CVD or type 2 diabetes (T2D) [4,5,6,7,8]. The high concentration of bioactive compounds, such as (poly)phenols, present in this dietary pattern may exert antioxidant, anti-inflammatory, and antithrombotic effects that contribute to the delay of CVD onset and progression [9,10,11].

Phenolic compounds are secondary plant metabolites contained in a large diversity of vegetables, fruits, nuts, seeds, cereals, oils, coffee, chocolate, and spices, and their structure has in common at least one aromatic ring attached to a hydroxyl group. These compounds show a wide structural diversity, and they can be grouped into five classes: flavonoids, phenolic acids, stilbenes, lignans, and other (poly)phenols [12]. In plants, phenolic compounds promote plant survival by acting as pollinators, protecting against ultraviolet radiation, and scavenging reactive oxygen species (ROS). Some of these protective features apparently translate to human health as well, promoting longevity by lowering the incidence of chronic diseases. The absorption, transport, bioavailability, and bioactivity of (poly)phenols after food intake vary depending on the type of compound. Some of them are absorbed in the stomach, while others, such as catechins, flavanols, and flavones, pass into the circulatory system through the small intestine. In general, the absorption of phenolic compounds is related to the release of aglycone, which is partly the result of microbial activity and digestive enzymes. During the aglycone route along the human body up to urine excretion, phenolic compound interactions may modulate potential risk factors of chronic disease by promoting cardioprotective effects and improving insulin sensitivity [13,14].

In addition, phenolic compounds may have antioxidant effects by counteracting senescence and protecting the mitochondrial function of vascular and endothelial smooth muscle cells by modulating signal transduction [15]. An increased expression of ROS-generating molecules and/or a decrease in antioxidant enzyme expression, such as catalase, superoxide dismutase, and glutathione peroxidase, promote the accumulation of ROS, causing damage to DNA, lipids, and proteins generating oxidative stress in the organism [16]. This accumulation of damaged molecules and dysregulation of mitochondrial function results in a state of oxidative stress, involving lipid peroxidation, which is associated with an increased incidence of age-related diseases, such as CVD, cancer, and other chronic diseases [17].

In this regard, oxidative modifications of low-density lipoproteins to oxidized LDL are one of the first events in the progression of atherosclerosis, and (poly)phenols have been shown to fight this process, exerting antioxidant activity through endogenous and exogenous mechanisms [18]. Flavonoids (flavonols, flavanols, flavanones, and flavones) and stilbenes are oxidized by free radicals, giving rise to more stable and less reactive compounds. Moreover, flavanones, anthocyanins, and phenolic acids have been shown to have antioxidant activity by enhancing cellular antioxidant defenses through activation of antioxidant and cytoprotective enzyme transcription factors [19,20]. Another pathway by which (poly)phenols exhibit their antioxidant function is modification of signal transduction that can promote up-regulation of antioxidant genes and induction of endogenous antioxidant enzymes, such as glutathione peroxidase, superoxide dismutase, catalase, or glutathione reductase [18].

Many sources of phenolic compounds, such as red wine, extra-virgin olive oil, and green tea, have the potential to improve vascular health by promoting the production of vasoprotective factors and by enhancing vascular smooth muscle function. Endothelial dysfunction is greatly caused by reduced nitric oxide (NO) availability because of increased oxidative stress, free radical generation, and other stress factors; (poly)phenols can improve NO release from endothelial cells, leading to activation of cyclic guanosine monophosphate in vascular smooth muscle cells and exerting a vessel relaxation, antioxidant, anti-inflammatory, and antithrombotic effect [18,21]. The antioxidant effect of (poly)phenols probably reflects reduced oxidative stress and may prevent vasoconstriction and pro-inflammatory responses.

In contrast, some negative effects have been reported for certain phenolic compound intake, e.g., iron chelation. The inhibiting effect of (poly)phenols on iron absorption has been shown in black tea and herb teas and may lead to poor iron status [22].

The purported benefits of (poly)phenol intake (PPI) are best evaluated in prospective cohorts with long follow-up, good control for confounding, a high retention rate, and a high variability in PPI. In the traditional Mediterranean diet, a high intake of phenolic compounds through olives, extra-virgin olive oil, and red wine may play an important role in cardiovascular prevention. These sources of phenolic compounds are likely to be different from those found in non-Mediterranean countries. However, when average intake is high, as is expected to happen with the Mediterranean diet, wide intake variability can be found. The aim of this study was to evaluate prospectively in a Mediterranean population the association of total PPI, and (poly)phenol subclasses with the incidence of CVD in the Seguimiento Universidad de Navarra (SUN) cohort of university graduates.

## 2. Materials and Methods

### 2.1. Study Population

The SUN project is a prospective, multipurpose, and dynamic cohort of Spanish university graduates whose recruitment started in December 1999 and nowadays involves almost 23,000 Spanish college graduates. A more detailed description of the cohort design and methodology is available elsewhere [23]. Briefly, self-reported questionnaires received through postal mail or the web collected information at baseline about sociodemographic characteristics, health status, family history of diseases, and lifestyle of the participants. Every two years, information about lifestyle factors, especially eating habits and health outcomes, was updated.

For these analyses, participants who were recently recruited and had less than 2 years and 9 months of follow-up (*n* = 341) were excluded, as well as those whose total energy intake was outside of predefined limits (<500 or >3500 kcal/d in females, or <800 or >4000 kcal/d in males) [24] (*n* = 2142) and who did not adequately answer the food frequency questionnaire (FFQ) (more than 15 items missing) (*n* = 2552). Additionally, participants with prevalent CVD (*n* = 247) and lost follow-up (*n* = 1465) were excluded. In total, 16,147 participants were included in this analysis, with a mean follow-up time of 11.5 years (SD = 4.5) and a retention rate of 91.7% (Figure 1).

### 2.2. Assessment of Dietary Intake and (Poly)Phenol Intake

A validated semi-quantitative FFQ with 136 food records was used to assess previous dietary intake and after a 10-year follow-up [25,26]. Each food was grouped into nine categories of consumption (never or seldom; 1–3 times/month; once a week; 2–4 times/week; 5–6 times/week; once a day; 2–3 times/day; 4–6 times/day and >6 times/day). To compute daily energy and nutrient intakes, Spanish food composition tables were used, and standard serving sizes were multiplied by the frequency of consumption of each food [27,28].

(Poly)phenol content provided by foods was obtained from the Phenol-Explorer database 3.6 version (www.phenol-explorer.eu, accessed on 1 October 2021) whose methodology is published elsewhere [29,30]. Foods with no or only traces of (poly)phenols were excluded. For the recipes and processed foods, (poly)phenol content was computed according to their ingredients. The most common extraction method for estimating PPI was high-performance liquid chromatography (HPLC), which quantifies (poly)phenol glycosides, phenolic acid esters together with aglycones and free phenolic acids concurrently, and normal-phase HPLC for proanthocyanidins. In the case of lignans, and phenolic acids in certain foods, such as cereals, olives, and beans, data corresponding to HPLC after hydrolysis were used to calculate the aglycone. This analytical method uses acid or alkaline hydrolysis to release certain compounds, such as insoluble phenolic acids esterified to the cell wall of cereals, which cannot be solubilized without hydrolysis. When food items of the FFQ included more than one food, a weighted average was used according to the average dietary intake in the adult Spanish population [31]. Finally, the individual PPI of each food was obtained by multiplying the content of each (poly)phenol by the daily consumption of each food. The total PPI was determined (milligrams per day) as the sum of all the individual PPIs of every food record reported by the FFQ. A ratio of the total or individual (poly)phenols contributed by the specific food or food group to the total (poly)phenol intake of all foods was calculated.

### 2.3. Outcome Assessment

The incidence of CVD was the endpoint of this study, and it included myocardial infarction diagnosed using universal criteria, non-fatal stroke defined as a sudden onset focal-neurological deficit with a vascular mechanism that lasted more than 24 h, and CVD death [32]. Clinical records or information were requested to participants or their families to confirm the event when CVD was self-reported in any of the biennial follow-up questionnaires. All CVD case information was revised and confirmed by an expert medical panel that was blind to participant diet and lifestyle exposure and classified according to the International Classification of Diseases (ICD 10). CVD deaths were confirmed through death certificates and the database of the Spanish National Statistic Institute.

### 2.4. Assessment of Covariates

At baseline, participants provided information about possible confounders, including socio-demographic characteristics (sex, age, level of education, and marital status), validated anthropometric measures [33] (weight and height), and health-related behavior such as alcohol intake, smoking status, leisure-time watching TV, or physical activity (METs-h/week) [34]. Family and personal medical history (hypertension, hypercholesterolemia, diabetes, cancer, or CVD, among others) was also reported and ascertained at baseline. Some validation studies in the SUN cohort assessed the reliability of self-reported medical data [35,36].

### 2.5. Statistical Analysis

PPI values were adjusted for total energy according to the residual method proposed by Willett [24] and were categorized into sex-specific quintiles. Descriptive statistics were used to analyze participants’ characteristics at baseline adjusted for age and sex using the inverse probability weighting method, using proportions or means (and standard deviations, SD) according to quintiles of PPI. The main sources of total PPI, and (poly)phenol classes and subclasses intake were calculated, as well as the contribution of each food item to total (poly)phenol and subclasses intake.

Crude and multivariable Cox regression models were used to analyze CVD risk and PPI using age as the underlying time variable, with quintiles of total energy-adjusted PPI intake or PPI classes as the independent variable. Time at entry was defined as the date of completion of the first questionnaire, and time at exit as the date of the cardiovascular event or date at which participants filled out their last follow-up questionnaire or the date of death due to a non-CVD-related cause for non-cases. Multivariable adjusted model 1 was stratified by age (five-year periods), recruitment period, marital status, and years of university education and adjusted for energy intake (kcal/day), smoking status (never smoker, current smoker or former smoker), lifetime tobacco exposure (packs-years), body mass index (BMI) (kg/m^2^) and the quadratic term, prevalent dyslipidemia, hypertension, and diabetes, family history of CVD, physical activity (METs-h/week, continuous), TV watching (hours/day, continuous), use of cardiovascular drugs (yes/no), health consciousness (number of medical check-ups, quintiles), energy-adjusted alcohol (g/day) intake, and energy-adjusted (g/day) sodium intake.

Hazard ratios (HRs) were calculated with their 95% confidence interval (CIs), including age as the underlying time variable and always considering the lowest quintile as the reference category. Repeated measurements were performed to evaluate dietary intake variation effect on those participants who completed the FFQ after 10-year follow-up (Q10 questionnaire) using cumulative average for dietary variables to reduce the effect of dietary intake variations at 10 years of follow-up, and the analysis was stratified and adjusted for the same confounders as in model 1. HRs in model 1 and repeated measurements were calculated excluding participants with 40 years of age or less to evaluate if the effect was restricted to subjects older than 40 years. The date on which they turn 40 years old, if applied, was used as entry in the regression model adjusted for the same confounders as the initial sample. In addition, HRs were estimated to assess the effect of a low (or suboptimal) PPI (first quintile of total PPI) versus a moderate-high intake (quintiles 2 to 5 of PPI). All analyses were repeated for each (poly)phenol subclass.

Multivariable-adjusted Nelson–Aalen curves were employed to describe the incidence of CVD across quintiles (Q1 vs. Q2–Q5) of total PPI to assess absolute risk along the follow-up period. To adjust these curves, inverse probability weighting methods were applied. Sensitivity analyses were performed by rerunning the same analysis, not excluding participants with incomplete FFQ at baseline, and the results remained similar. The analyses were performed with the Stata statistical software package version 16 (College Station, TX, USA; Stata Corp LLC). All *p*-values presented are two-tailed, and the statistical significance was set at 0.05.

## 3. Results

The main baseline characteristics of the 16,147 participants included in the analyses according to the energy-adjusted quintiles of the baseline total PPI are shown in Table 1. The mean age of the participants was 37 years (SD = 12), and the median BMI was 23.5 kg/m^2^ (SD = 3.5).

Participants in the first quintile of total PPI showed the highest total energy intake and fat intake, while carbohydrate, protein, fiber, and alcohol intakes were the lowest for the lowest quintile. This first quintile also exhibited the lowest average level of leisure-time physical activity (MET-h/week) at baseline. Participants in the lowest quintile of PPI showed a higher use of cardiovascular drugs, hypertension, and diabetes at baseline but also the highest proportion of never smokers. Flavonoid intake was the (poly)phenol class that increased most across quintiles of total PPI. Adherence to the Mediterranean diet increased progressively from the first quintile to the fifth.

The energy-adjusted mean ± SD of total PPI was 779.7 ± 371.6 mg/day, of which 55.5% were flavonoids (432.8 ± 271.6 mg/day), 38.8% phenolic acids (302.9 ± 151.9 mg/day), 5% other (poly)phenols (40.5 ± 33.1), 0.28% lignans (2.2 ± 1.0 mg/day), and 0.15% stilbenes (1.2 ± 2.6 mg/day) (Table 2). The class that contributed the most to total PPI was flavonoids, followed by phenols acids, but the highest subclass contributor to total PPI was a phenolic acids subclass, hydroxycinnamic acids (266.3 ± 140.1 mg/day, 34.15%), followed by a flavonoid subclass, proanthocyanidins (221.3 ± 198.2 mg/day, 28.38%).

Regarding (poly)phenol subclasses, the main contributor to flavonoids was proanthocyanidins (221.3 ± 198.2 mg/day) and the principal food source of flavanoids was chocolate (44.4%), apple (21%), cherries (11.1%), and strawberries (5.3%). The group of flavanones (74.6 ± 75.8 mg/day) was the second subclass contributor to flavonoids and was basically provided by oranges (33.7%), natural orange juice (27.5%), and fruit juices from concentrates (18.7%). Among phenol acids, hydroxycinnamic acids had the highest intake (66.3 ± 140.1 mg/day) with coffee (31.8%) and decaffeinated coffee (12.7%) as the main food sources, and hydroxybenzoic acids showed lower intakes (31.6 ± 24.8 mg/day) largely provided by nuts (34%), olives (16.41%), and strawberries (13%). Olives were the main food source for hydroxyphenylacetic acids (4.44 ± 6.9 mg/day) and hydroxyphenylpropanoic acids (0.6 ± 1 mg/day). Lignan intake (2.2 ± 1.1 mg/day) was mostly provided by carrot and pumpkin (13.3%) and olive oil (11.2%), while the principal supply for stilbenes (1.2 ± 2.6 mg/day) was red wine (88.6%). Tyrosol intake (27.2 ± 29.9 mg/day) was the most abundant element within the subclass of “other (poly)phenols” with olives (66.1%) and olive oil (28.9%) as principal food contributors.

Among 185,652 person-years of follow-up, a total of 123 medically-confirmed new onset cases of CVD were observed (median follow-up 11.5 years). A low intake vs. medium–high total PPI analysis showed a significantly higher risk of CVD incidence (adjusted multivariable model 1 HR_Q1 vs. Q2–Q5_ 2.12; 95% CI 1.27–3.57; *p* < 0.001) (Table 3). When analyses were restricted to participants older than 40 years, lower HRs were observed, but results did not remain significant. Once repeated measurements were applied, an increased risk remained significant (adjusted multivariable model 1 HR_Q1 vs. Q2–Q5_ 1.85; 95% CI 1.09–3.16; *p* < 0.05).

When low versus medium–high flavonoid intake was analyzed, the results showed a significantly higher risk in quintile 1 of flavonoid intake in the crude model. (HR_Q1 vs. Q2–Q5_ 1.86; 95% CI 1.20–2.87; *p* < 0.001), multivariable model 1 (HR_Q1 vs. Q2–Q5_ 1.73; 95% CI 1.03–2.90; *p* < 0.05) and using repeated measurements (HR_Q1 vs. Q2–Q5_ 1.88; 95% CI 1.11–3.17; *p* < 0.05). Restricting the analyses to participants older than 40 years, the results remained significant, and HRs were higher in multivariable model 1 (HR_Q1 vs. Q2–Q5_ 2.27; 95% CI 1.33–3.88; *p* < 0.001) and in repeated measurement analyses (HR_Q1 vs. Q2–Q5_ 1.97; 95% CI 1.19–3.27; *p* < 0.001).

For lignan intake, significant results were only observed in a low intake when analyses were restricted to participants older than 40 years in multivariable model 1 (HR_Q1 vs. Q2–Q5_ 1.83; 95% CI 1.03–3.25; *p* < 0.05), but it was the only class of phenolic compounds that showed a significant linear trend (*p* < 0.05) across quintiles in all regression models conducted. Regarding phenol acids, significant results were only observed for low intake (HR_Q1 vs. Q2–Q5_ 1.97; 95% CI 1.12–3.45; *p* < 0.05) when analyses were conducted on participants older than 40 years in multivariable model 1.

Cumulative HR for CVD development over time across quintiles of total PPI was graphically represented using Nelson–Aalen curves with adjustment for the same variables as model 1 but using inverse probability weighting and according to a low total PPI (first quintile, Q1) versus medium–high total PPI (Q2–Q5), which showed a higher absolute risk of CVD in the lowest quintile of total PI, with diverging curves starting after 2 year follow-up (Figure 2). Sensitivity analyses were performed repeating the same analysis without excluding the 2552 participants with incomplete questionnaires (more than 15 items missing) at baseline), and the results remained similar.

## 4. Discussion

These results suggest that a suboptimal intake (432.1 mg/day or less) of total (poly)phenols and a low flavonoid intake (190.7 mg/day or less) are associated with increased CVD risk. These findings are in line with previous analyses of PPI in a smaller subsample of the SUN cohort, with a shorter follow-up and in which repeated measurements of PPI during follow-up were not used [38]. It can now be confirmed that the cardioprotective effect persists when repeated measurements at 10 years of follow-up are performed. In addition, the natural history of the occurrence of CVD related to phenolic compounds could be plotted, showing changes in absolute risks over the follow-up period.

Evidence from randomized controlled trials designed to assess primary prevention of cardiovascular disease in Mediterranean populations described the role of phenolic compounds in cardiovascular health and reported an inverse relationship between dietary intake of phenolic compounds and the risk of CVD, consistent with the results shown in this study [39,40]. Findings also revealed a meaningful role of flavonoid subclasses in reducing the risk of CVD. They could be explained by proanthocyanidins and flavonols intake, which are the main flavonoid compounds contributing to the total PPI. Some studies supported a beneficial effect of proanthocyanidin-rich foods, but they also mentioned that bioavailability of the polymerized proanthocyanidins is very limited, and the biological active compounds should be investigated in the monomeric forms of the proanthocyanidins, the catechins, and in the set of metabolic compounds generated by degradation of the proanthocyanidins in the intestine, which can be interrelated with gut microbiota activity [41,42,43,44].

A dose-response meta-analysis [45] studied the association between dietary intake of total subclasses and individual flavonoids and risk of CVD. It provided evidence about the potential cardiovascular benefits of a flavonol-rich diet. Moreover, a systematic review and meta-analysis of prospective cohort studies found that flavonol intake was inversely associated with CVD risk in either US or European populations [46]. In addition, the results of the dose-response analysis indicated that an average increase of 10 mg/day of flavonol intake was associated with a 5% lower risk of CVD. Possible mechanisms by which flavonols may decrease the risk of CVD probably involve more than one effect. In brief, there is solid evidence that, in vitro, quercetin, and related flavonols exert endothelium vasodilator effects, protective effects on endothelial function against oxidative stress, platelet antiaggregant effects, inhibition of low-density lipoprotein oxidation, and reduction of adhesion molecules and other inflammatory markers [47,48]. Moreover, evidence suggests that quercetin may be a key compound preventing the most common forms of CVD and contributing to the protective effects provided by fruits and vegetables, which is consistent with food sources of flavonols reported here in the SUN cohort [49,50].

Regarding lignans, a significantly higher risk of CVD was found in the lowest intake quintile and a significant inverse linear trend across quintiles of intake in participants older than 40 years. Previous studies found a lower risk of total CVD associated with a high long-term intake of lignans, although the effect may be explained by a possible synergy with fiber intake [51]. Lignans in the present study were mainly supplied by carrot and pumpkin (13.3%), which are vegetables with high concentrations of beta-carotenes. In this context, a review indicated that beta-carotene and its vitamin A derivatives may stimulate lipid catabolism in several tissues and may play a role in immune cells, which are related to the physiopathology of CVD onset [52,53]. Olive oil was the second source of lignans (11.2%), and its protective role against CVD has been widely studied both in Mediterranean and non-Mediterranean countries [7,54,55,56,57].

Phenolic acid intake, of which coffee hydroxycinnamic acids were the main source in this study, showed a significant increase of CVD risk in the lowest quintile of intake in those over 40 years of age, and some evidence explained the mechanisms of endothelial protection by hydroxycinnamic acids, suggesting a potential role in prevention of thrombus formation or atherosclerotic lesion development [58]. Recent data from large observational studies have shown possible associations between phenolic acid intake and improvements in blood pressure and metabolic syndrome, although evidence from randomized clinical trials presents some inconsistencies due to individual variation in the diversity of the microbiota and the influence it exerts on the effective use of phenolic acids in the human body [59].

Randomized controlled trials have evaluated the role of isolated phenolic compounds in cardiovascular prevention, taken as a complement and eliminating the possible synergies that could be obtained by the contribution of nutrients, vitamins, and minerals from fruits and vegetables that are naturally present and that can be a bias when evaluating the cardioprotective effect. In some of them, improvements in endothelial function, metabolic syndrome components, lipid metabolism parameters, and antioxidant and anti-inflammatory improvement were shown, but in contrast, another randomized controlled trial with a daily intake of 25 mg pure (-)-epicatechin for 2 weeks did not show cardiometabolic risk factor improvements outside of its food matrix [60,61,62,63]. Therefore, it appears that the action of polyphenols on human metabolism may change when ingested in isolation or in combination with the dietary components of food.

Some limitations may be observed in the present study. One of them can be related to the estimation of PPI, even though the most updated and comprehensive database available was used (Phenol-explorer version 3.6), not all foods from FFQ are included in the database (e.g., honey), FFQ does not cover all (poly)phenol-rich foods (e.g., spices or herbs) and also does not distinguish food details that can affect the (poly)phenol content (types of tea, coffee or cocoa percentage of different varieties of chocolate) nonetheless some validation studies have shown that FFQ are soundly valid for estimating (poly)phenol intake [64]. Furthermore, bioavailability after ingestion was not considered and may vary among individuals. In addition, phenolic intake can be affected by external conditions that may affect the food (poly)phenol content, such as ripeness, processing, and storage, and variety can affect the food (poly)phenol richness [12]. Fortunately for our participants, the number of events in the SUN project was relatively low, as expected in a relatively young and healthy Mediterranean cohort. Nevertheless, this fact may have limited our statistical power for assessing these associations, and the lack of significance in HR of quintiles may be due to the low number of events in this Mediterranean cohort.

This study also has several strengths, including its prospective design, comprehensive, and validated data about potential confounders, ascertainment of events by a medical panel, and a high retention rate of the cohort (92%). Additionally, dietary measurement errors were reduced by considering changes in diet at the 10-year follow-up, which enabled us to calculate the cumulative average of nutrient exposure for a better representativeness of dietary intake over time. Moreover, this study was conducted in a Mediterranean country, which may help to adequately capture the effect of high (poly)phenol content foods, such as olive oil or red wine traditionally present in the Mediterranean diet and with a high observed consumption in this population.

## 5. Conclusions

These findings support the avoidance of suboptimal PPI and the promotion of (poly)phenol-rich diets, especially flavonoid-rich dietary patterns, to reduce the risk of CVD. Flavonoids were found most abundantly in fruits, vegetables, cocoa, and wine in our cohort. Future research, especially randomized controlled trials, should be led to control phenolics intake confounding by culinary preparations that strongly influence the bioavailability of these compounds. Further research is needed to characterize interactions with gut microbiota and with systemic tissues.

## Figures and Tables

**Figure 1 antioxidants-11-00783-f001:**
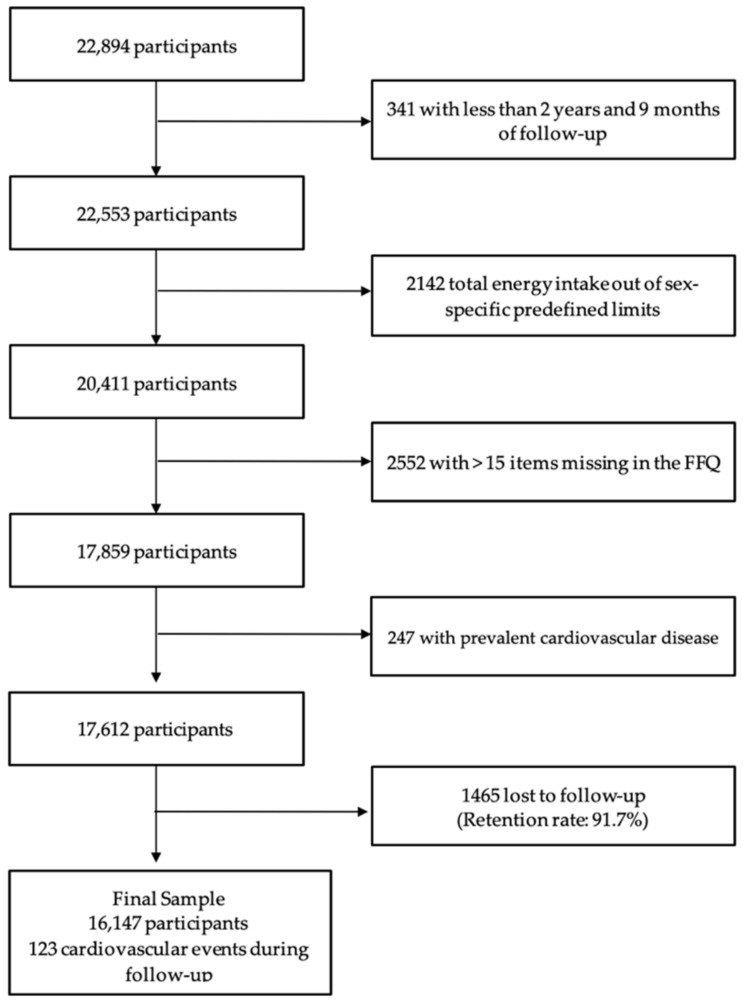
Flowchart of participants in the SUN (‘Seguimiento Universidad de Navarra’) included in analyses of (poly)phenol intake and incident cardiovascular diseases. Abbreviation: Food frequency questionnaire (FFQ).

**Figure 2 antioxidants-11-00783-f002:**
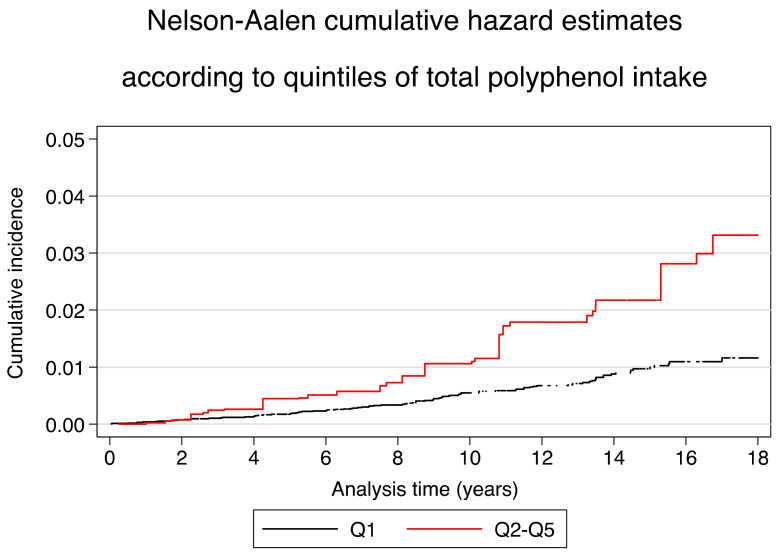
Nelson–Aalen cumulative hazard ratios estimates of cardiovascular disease according to low intake (Q1) vs. a medium-high (Q2–Q5) total PI. Multivariable adjusted model 1 using inverse probability weighting to adjust for sex, age, energy intake (kcal/day), smoking status (never smoker, current smoker or former smoker), lifetime tobacco exposure (packs-years), BMI (kg/m^2^) and the quadratic term, dyslipidemia(yes/no), hypertension(yes/no), diabetes, family history of CVD (yes/no), physical activity (metabolic equivalents-h/week), TV watching (hours/day), use of cardiovascular drugs (yes/no), health consciousness (quintiles), energy-adjusted alcohol intake (g/day), and energy-adjusted sodium intake (g/day).

**Table 1 antioxidants-11-00783-t001:** Baseline characteristics * of participants across sex-specific energy-adjusted quintiles of total (poly)phenol dietary intake.

	Energy-Adjusted Quintiles of Total (Poly)Phenol Intake				
	Q1 (*n* = 3230)	Q2 (*n* = 3229)	Q3 (*n* = 3230)	Q4 (*n* = 3229)	Q5 (*n* = 3229)
Age (years)	38.5 (13.5)	37.3 (12.0)	37.1 (11.5)	37.2 (11.2)	37.2 (11.6)
Sex (% female)	61.0%	61.1%	61.3%	61.4%	62.2%
Total (poly)phenol intake (mg/day)	375.7 (90.7)	547.1 (36.5)	667.36 (35.6)	806.4 (49.1)	1149 (304.4)
Flavonoids intake (mg/day)	202.9 (85.7)	313.1 (78.4)	394.4 (91.5)	495.4 (116.4)	768.9 (312.6)
Phenol acids intake (mg/day)	180.7 (70.0)	250.0 (74.5)	294.0 (85.8)	343.9 (107.2)	441.8 (180.3)
Lignans intake (mg/day)	1.6 (0.6)	1.9 (0.6)	2.2 (0.7)	2.4 (0.8)	2.9 (1.3)
Stilbenes intake (mg/day)	0.6 (1.1)	0.9 (1.6)	1.2 (2.1)	1.4 (2.6)	1.8 (3.7)
Other (poly)phenols intake (mg/day)	27.1 (19.2)	34.2 (19.6)	39.9 (22.2)	45.1 (27.5)	55.1 (50.6)
Body mass Index (kg/m^2^)	23.5 (3.6)	23.5 (3.5)	23.5 (3.5)	23.5 (3.5)	23.3 (3.6)
Physical activity (METs-h/week)	18.7 (21.5)	19.6 (20.3)	21.7 (22.2)	23.5 (23.4)	26.5 (27.8)
Marital status (% married)	37.9%	45.2%	51.9%	54.5%	57.9%
University education (years)	5.0 (1.4)	5.1 (1.5)	5.1 (1.5)	5.1 (1.5)	5.0 (1.5)
TV watching time (hours)	1.7 (1.3)	1.6 (1.2)	1.6 (1.2)	1.6 (1.2)	1.6 (1.2)
Smoking					
Current smoker	22.2%	22.0%	20.9%	22.4%	21.5%
Former smoker	26.0%	26.9%	29.7%	28.8%	29.6%
Never smoker	51.1%	50.4%	48.8%	48.1%	48.3%
Package-years of smoking	5.2 (11.1)	4.4 (9.2)	4.4 (8.9)	4.5 (8.7)	4.8 (9.3)
Dyslipidemia at baseline	15.1%	15.1%	16.8%	17.9%	17.8%
Hypertension at baseline	10.6%	10.2%	9.7%	9.2%	9.4%
Family history of CVD	7.7%	8.2%	7.1%	7.4%	8.2%
Diabetes at baseline	2.4%	1.1%	1.5%	2.1%	1.6%
Use of cardiovascular drugs	2.9%	2.6%	2.5%	2.3%	2.6%
Total energy intake (kcal/day)	2580 (563)	2310 (585)	2258 (595)	2305 (588)	2480 (609)
Carbohydrate intake (% energy)	42.4 (7.3)	42.2 (6.7)	42.8 (6.6)	43.9 (6.6)	46.0 (7.4)
Protein intake (% energy)	17.9 (3.2)	18.4 (3.0)	18.5 (3.0)	18.4 (3.2)	17.6 (3.0)
Fat intake (% energy)	38.2 (6.4)	37.6 (5.9)	36.7 (5.8)	35.6 (5.9)	34.2 (6.5)
Dietary fiber intake (g/day)	23.7 (8.8)	24.3 (9.0)	26.5 (9.4)	29.5 (10.6)	36.6 (15.2)
Alcohol intake (g/day)	5.5 (9.8)	5.9 (8.4)	6.3 (8.8)	7.0 (9.8)	7.5 (11.4)
Adherence to MDS (0–9 score)	3.5 (1.6)	3.9 (1.6)	4.3 (1.7)	4.8 (1.7)	5.2 (1.7)

Abbreviatures: Q, Quintile; MET metabolic equivalents; SUN, Seguimiento Universidad de Navarra; CVD Cardiovascular disease; MDS, Mediterranean Diet Score proposed by Trichopoulou et al. [37]. * Adjusted for inverse probability weighting for sex and age. Values are expressed as means and standard deviations or percentages.

**Table 2 antioxidants-11-00783-t002:** Contribution of (poly)phenol subclasses to total (poly)phenol intake and dietary sources.

(Poly)phenol Classes and Subclasses	Mean (mg/d) ± SD, (%)	Food Sources * (% of Contribution)
*Flavonoids*	432.9 ± 271.6, (55.5)	
Anthocyanins	37.9 ± 43.3, (4.8)	Cherries (45.6), strawberries (16), red wine (13.7), grapes (13.6), olives (8), beans (1.2) milkshakes (1.1).
Chalcones	0.031 ± 0.006, (< 0.1)	Beer (100).
Dihydrochalcones	1.9 ± 2.6, (0.2)	Apples (100).
Dihydroflavonols	1.4 ± 3.2, (0.2)	Red wine (98.3), other wines (1.8).
Flavan-3-ols	21.1 ± 16, (2.7)	Apples (26.9), chocolate (9.9), red wine (14.6), peaches (10.2), cherries (6.7), grapes (5.7), strawberries (3), green beans (2.5), banana (2.5) lentils (1.26).
Flavanones	74.6 ± 75.8, (9.6)	Oranges (33.7), natural orange juice (27.5), fruit juices from concentrate (18.7), other fruit juices (1.5), tomato (1.14).
Flavones	17.6 ± 13.4, (2.2)	Other vegetables (36.5), natural orange juice (17.3), cookies (11), olives (6), fruit juices from concentrate (4.4), chocolate cookies (3.3), watermelon (2.9), industrial bakery (2.3) pastries (2.2), croquettes (1.8), peppers (1.8), pizza (1.4), cupcake (1.1) carrots and pumpkin (1).
Flavonols	57.1 ± 35.8, (7.3)	Lettuce (36.5), swiss chard leaves (29.2), asparagus (7), olives (2), nuts (3.2), green beans (2.8), chocolate (1.7), cabbage (1.7), tomato (1.6), apples (1.5).
Isoflavonoids	0.03 ± 0.03, (< 0.1)	Beans (67.9), nuts (29.5), beer (2.5).
Proanthocyanidins	221.3 ± 198.2, (28.4)	Chocolate (44.4), apple (21), cherries (11.1), strawberries (5.34), grapes (4.7), nuts (4.5), red wine (3.1), beans (3).
*Lignans*	2.2 ± 1.1, (0.3)	Carrot and pumpkin (13.3), olive oil (11.2), tomato (9), broccoli and cabbage (7.6), oranges (6.8), green beans (6), peaches (4.4), pepper (3.5), strawberries (3.2), cookies (3) asparagus (2.8), gazpacho (2.8), red wine (2.7), apples (2.3), grapes (1.9), eggplant zucchini and cucumber (1.7), dried fruit (1.6), kiwi (1.4), fried potatoes (1.3), melon (1.3), nuts (1.24) bread (1.1).
*Phenolic acids*	302.9 ± 151.9, (38.8)	
Hydroxybenzoic acids	31.6 ± 24.8, (4.1)	Nuts (34.4), olives (16.4), strawberries (13), carrots and pumpkin (11.9), swiss chard leaves (7.4), red wine (5.2), beer (2.4), apples (1.4), banana (1).
Hydroxycinnamic acids	266.3 ± 140.1, (34.2)	Coffee (31.9), decaffeinated coffee (12.7), other vegetable (8.5) carrots and pumpkin (7.8) cherries (7), French fries (5.7), olives (4.9), apple (4.3), baked or boiled potatoes (3) tomato (1.85) nuts (1.6), peaches (1.5) orange juice (1.3) red wine (1.2).
Hydroxyphenylpropanoic acids	0.6 ± 1, (< 0.1)	Olives (100).
Hydroxyphenylacetic acids	4.44 ± 6.9, (0.6)	Olives (96.6), red wine (2.5).
*Stilbenes*	1.21 ± 2.6, (0.2)	Red wine (88.6), grapes (4), other wines (3.6), strawberries (2.5).
*Other (poly)phenols*	40.5 ± 33.14, (5.2)	
Alkylmethoxyphenols	0.3 ± 0.6, (0.3)	Decaffeinated coffee (84.6) beer (15.3).
Alkylphenols	8.1 ±11.2, (1)	Breakfast cereals (43.2), whole-grain bread (41.7), pasta (8), cookies (2.5).
Furanocoumarins	0.4 ± 0.8, (0.4)	Other vegetables (100).
Hydroxybenzaldehydes	0.2 ± 0.4, (0.2)	Red wine (90.7), beer (2.7), other wines (2.7), olives (2), whisky (1.4).
Hydroxybenzoketones	0.001 ± 0.002, (< 0.1)	Beer (100)
Hydroxycoumarins	0.04 ± 0.08, (< 0.1)	Beer (75), other wines (24.9)
Methoxyphenols	0.03 ± 0.07, (< 0.1)	Decaffeinated coffee (100)
Tyrosols	27.2 ± 29.9, (3.5)	Olives (66.1), olive oil (28.9), red wine (0.9)
Other (poly)phenols (subclass)	4.5 ± 6.4, (0.6)	Orange juice (65.2), other fruits juice (24.4), Coffee (5.61), apples (256), olives (1.5)

* Food sources that contribute more than 1%.

**Table 3 antioxidants-11-00783-t003:** Hazard ratio (HR) and 95% confidence intervals (CI) of confirmed cardiovascular disease cases according to quintiles of total (poly)phenol intake and (poly)phenol subclasses.

	Energy-Adjusted Quintiles of Total (Poly)Phenol Intake						Low Intake vs. Medium–High Intake
	1	2	3	4	5	*p* For Trend	
	*n* = 3230	*n* = 3229	*n* = 3230	*n* = 3229	*n* = 3229		Q1 vs. Q2–Q5
**Total (poly)phenol intake**							
Median intake (mg/d)	432.1	600.4	729.7	883.4	1171.1		
Cases	27	16	18	26	36		
Person-years	37038	37880	37610	36657	36468		
Age-sex adjusted HR (95% CI)	1 (ref.)	0.42 (0.22–0.78) **	0.43 (0.24–0.79) **	0.52 (0.30–0.9) *	0.56 (0.33–0.93) *	0.29	2.04 (1.32–3.15) **
Multivariable adjusted model 1	1 (ref.)	0.33 (0.15–0.71) **	0.47 (0.23–0.95) *	0.55 (0.29–1.05)	0.52 (0.28–0.96) *	0.29	2.12 (1.27–3.57) **
Model 1 restricted to >40 years	1 (ref.)	0.55 (0.27–1.14)	0.66 (0.33–1.32)	0.64 (0.33–1.24)	0.66 (0.35–1.24)	0.34	1.58 (0.94–2.64)
Repeated Measurements adjusted model 1	1 (ref.)	0.39 (0.18–0.89) *	0.57 (0.28–1.14)	0.63 (0.33–1.21)	0.56 (0.30–1.05)	0.30	1.85 (1.09–3.16) *
Model 1 restricted to >40 years	1 (ref.)	0.56 (0.27–1.16)	0.67 (0.34–1.31)	0.68 (0.34–1.20)	0.64 (0.34–1.20)	0.35	1.56 (0.93–2.60)
**Flavonoids intake**							
Median intake (mg/d)	190.7	300.3	385.7	496.2	720.4		
Cases	27	13	29	24	30		
Person-years	37701	37642	37449	36618	36243		
Age-sex adjusted HR (95% CI)	1 (ref.)	0.35 (0.18–0.69) **	0.70 (0.41–1.19)	0.54 (0.31–0.94) *	0.54 (0.32–0.92) *	0.19	1.86 (1.20–2.87) **
Multivariable adjusted model 1	1 (ref.)	0.32 (0.14–0.71) **	1.01 (0.54–1.89)	0.59 (0.31–1.15)	0.51 (0.26–1.00) *	0.13	1.73 (1.03–2.90) *
Model 1 restricted to >40 years	1 (ref.)	0.27 (0.12–0.63) **	1.11 (0.60–2.04)	0.52 (0.26–1.03)	0.48 (0.24–0.94) *	0.10	2.27 (1.33–3.88) **
Repeated Measurements adjusted model 1	1 (ref.)	0.27 (0.11–0.63) **	0.88 (0.47–1.65)	0.64 (0.33–1.22)	0.44 (0.22–0.88) *	0.08	1.88 (1.11–3.17) *
Model 1 restricted to >40 years	1 (ref.)	0.28 (0.14–0.64) **	0.85 (0.46–1.58)	0.51 (0.26–1.00)	0.44 (0.22–0.87) *	0.06	1.97 (1.19–3.27) **
**Lignans intake**							
Median intake (mg/d)	1.2	1.7	2.1	2.5	3.3		
Cases	26	32	21	23	26		
Person-years	38692	38276	37372	36279	35034		
Age-sex adjusted HR (95% CI)	1 (ref.)	1.10 (0.63–1.92)	0.73 (0.40–1.35)	0.66 (0.36–1.21)	0.61 (0.33–1.11)	0.03	1.29 (0.78–2.09)
Multivariable adjusted model 1	1 (ref.)	1.12 (0.60–2.09)	0.72 (0.35–1.45)	0.75 (0.37–1.52)	0.51 (0.25–1.04)	0.02	1.26 (0.73–2.19)
Model 1 restricted to >40 years	1 (ref.)	0.95 (0.51–1.78)	0.79 (0.38–1.50)	0.65 (0.32–1.31)	0.42 (0.20–0.87) *	<0.01	1.83 (1.03–3.25) *
Repeated Measurements adjusted model 1	1 (ref.)	1.41 (0.74–2.68)	0.80 (0.39–1.64)	0.87 (0.42–1.82)	0.61 (0.30–1.29)	0.04	1.06 (0.59–1.90)
Model 1 restricted to >40 years	1 (ref.)	1.40 (0.74–2.60)	0.87 (0.43–1.73)	0.72 (0.33–1.53)	0.58 (0.28–1.24)	0.03	1.08 (0.62–1.89)
**Phenolic acids intake**							
Median intake (mg/d)	146.3	220.7	285.1	351.2	480.3		
Cases	20	27	19	22	35		
Person-years	36959	36959	37068	37682	37486		
Age-sex adjusted HR (95% CI)	1 (ref.)	0.82 (0.46–1.47)	0.54 (0.29–1.02)	0.62 (0.33–1.15)	0.78 (0.44–1.37)	0.55	1.44 (0.89–2.35)
Multivariable adjusted model 1	1 (ref.)	0.70 (0.35–1.40)	0.35 (0.16–0.80) *	0.57 (0.28–1.15)	0.67 (0.34–1.29)	0.48	1.74 (0.99–3.09)
Model 1 restricted to >40 years	1 (ref.)	0.74 (0.38–1.43) *	0.39 (0.20–0.86) *	0.62 (0.31–1.24)	0.78 (0.42–1.44)	0.53	1.97 (1.12–3.45) *
Repeated Measurements adjusted model 1	1 (ref.)	0.92 (0.46–1.82)	0.44 (0.20–0.95) *	0.60 (0.29–1.25)	0.77 (0.39–1.52)	0.50	1.47 (0.82–2.62)
Model 1 restricted to >40 years	1 (ref.)	0.62 (0.32–1.23)	0.43 (0.20–0.91) *	0.60 (0.30–1.19)	0.71 (0.4–1.33)	0.57	1.66 (0.98–2.80)
**Stilbenes intake**							
Median intake (mg/d)	−0.01	0.1	0.4	0.8	3.5		
Cases	16	10	23	22	52		
Person-years	37629	36446	36290	36961	38328		
Age-sex adjusted HR (95% CI)	1 (ref.)	0.61 (0.27–1.34)	1.05 (0.55–1.99)	0.8(0 0.42–1.54)	1.26(0.71–2.24)	0.05	1.01 (0.60–1.72)
Multivariable adjusted model 1	1 (ref.)	0.55 (0.22–1.38)	1.21 (0.55–2.64)	0.80 (0.36–1.75)	1.1(0.52–2.27)	0.44	1.11 (0.58–2.10)
Model 1 restricted to >40 years	1 (ref.)	0.72 (0.32–1.60)	0.82 (0.36–1.88)	0.91 (0.43–1.93)	1.05(0.51–2.19)	0.52	0.94 (0.46–1.90)
Repeated Measurements adjusted model 1	1 (ref.)	0.48 (0.19–1.19)	0.96 (0.44–2.10)	0.76 (0.35–1.63)	0.97 (0.47–2.00)	0.43	1.26(0.66–2.36)
Model 1 restricted to >40 years	1 (ref.)	0.94 (0.44–2.00)	0.55 (0.22–1.34)	0.82 (0.39–1.75)	1.12(0.54–2.32)	0.60	1.13(0.6–2.11)
**Other (poly)phenols intake**							
Median intake (mg/d)	14.2	25.4	34.0	45.2	72.1		
Cases	30	21	22	21	29		
Person-years	38332	37833	36938	36155	36396		
Age-sex adjusted HR (95% CI)	1(ref.)	0.70 (0.40–1.23)	0.80 (0.45–1.38)	0.74 (0.42–1.39)	0.79 (0.47–1.32)	0.57	1.32 (0.87–2.00)
Multivariable adjusted model 1	1(ref.)	0.87 (0.45–1.70)	0.89 (0.45–1.76)	0.76 (0.38–1.51)	0.95 (0.51–1.79)	0.92	1.15 (0.68–1.93)
Model 1 restricted to >40 years	1(ref.)	0.81 (0.41–1.59)	0.81 (0.40–1.62)	0.74 (0.37–1.50)	0.98 (0.52–1.87)	0.94	1.18 (0.66–2.08)
Repeated Measurements adjusted model 1	1(ref.)	0.74 (0.38–1.45)	0.77 (0.39–1.50)	0.79 (0.40–1.55)	0.91 (0.50–1.70)	0.97	1.23 (0.74–2.05)
Model 1 restricted to >40 years	1(ref.)	0.84 (0.42–1.62)	0.62 (0.31–1.26)	0.87 (0.45–1.75)	0.96 (0.51–1.82)	0.81	1.21 (0.72–2.02)

* *p* < 0.05, ** *p* < 0.01. Abbreviatures: CI, confidence interval; HR, hazard ratio; Q, quintile; ref., reference. All Cox regression models used age as the underlying time variable and were also stratified by age (five-year periods), recruitment period, marital status, and years of university education. Multivariable adjusted model 1: Additionally adjusted for energy intake (kcal/day), smoking status (never smoker, current smoker or former smoker), lifetime tobacco exposure (packs-years), BMI (kg/m^2^) and the quadratic term, dyslipidemia, hypertension (yes/no), diabetes(yes/no), family history of cardiovascular disease (yes/no), physical activity (metabolic equivalents-h/week), TV watching (hours/day), use of cardiovascular drugs (yes/no), health consciousness (quintiles), energy-adjusted alcohol intake (g/day) and energy-adjusted sodium intake (g/day). Repeated measurements model 1 were adjusted by the same variables on model 1 with updated data at 10 years of follow-up (except physical activity, TV watching, and health consciousness) using cumulative average for dietary variables.

## Data Availability

All of the data is contained within the article.

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
