# Peer review of "Effect of Dietary Phenolic Compounds on Incidence of Cardiovascular Disease in the SUN Project; 10 Years of Follow-Up"

_antioxidants, 2022, doi:10.3390/antiox11040783_

Round 1

Reviewer 1 Report

In the abstract we could find: Some studies have pointed out the benefits of plant-based diets and that the high concentration of phenolic compounds present in these dietary patterns may play an important role in cardiovascular prevention. 

It should be underlined in abstract that the main source of polyphenols in Spanish diet are olives , olive oil , red wine. This plant-based diets differ from typical diets in other countries 

Author Response

Response to Reviewer #1

In the abstract we could find: Some studies have pointed out the benefits of plant-based diets and that the high concentration of phenolic compounds present in these dietary patterns may play an important role in cardiovascular prevention. It should be underlined in the abstract that the main sources of polyphenols in Spanish diet are olives, olive oil, red wine. These plant-based diets differ from typical diets in other countries

The abstract has been modified to clarify this point. The new version now reads as: “Health benefits of plant-based diets have been reported. Plant-based diets found in Spain and other Mediterranean countries differ from typical diets in other countries. In the Mediterranean diet, a high intake of phenolic compounds through olives, olive oil and red wine may play an important role in cardiovascular prevention”.

Reviewer 2 Report

The work proposed to Antioxidants is entitled "Effect of dietary polyphenol intake on incidence of cardiovascular disease in the SUN project; 10 years of follow-up." The importance of this issue is huge for worldwide population inside the scenario of NCDs world heath problem. Some improvements are needed in this work, that are listed below. Also, a better abstract and conclusion should be provided. The introduction is poor and the discussion fell short of the expected for an Antioxidant publication. Some scheme or figure is lacking in the work presented. 

Please, correct all throughout the text the use of words such as "we", "our", and other personalisations, science should be written in the third person.
Could be included a sentence, in the end of abstract, regarding the impact of findings achieved in the work described.
Line 35 - What is "risk factors1", please correct finger typing mistakes.

The authors should clarify if they are talking about polyphenols or if they are talking about phenolic compounds. Even in the literature we find a confusion between the use of these two names, they are chemically different compounds. Please, verify what compound you are talking about, with chemical rigour. Given the definition " they can be classified into five classes, flavonoids, phenolic acids, stilbenes, lignans.." - The authors are chemically talking about phenolic compounds. Errors in science should not be continually spread. This correction is needed all over the text.
Line 53- "the final compound aglycone is absorbed and enter to the portal vein circulation reaching other tissues..." - This is not true for all phenolic compounds, most of them are not, or poorly absorbed, and do not reach other tissues. Please, use and cite more accurate references.
Line 55 - "During the aglycone route along human body up to urine excretion, phenolic compounds interactions may modulate potential risk factors of chronic diseases promoting cardioprotective effects and improving insulin sensitivity into five classes, flavonoids, phenolic acids, stilbenes, lignans, and other polyphenols into five classes, flavonoids, phenolic acids, stilbenes, lignans, and other polyphenols [13,14]. " - Please, correct this sentence.

Line 73 - "Also, consuming seeds, nuts or legumes rich in phytates could also impede iron absorption." - Please, insert the reference of this sentence. Or is it an achievement of the present work research that is mentioned in the introduction section?

Clearly, the introduction needs extensive improvement to achieve the standard of quality needed in the Antioxidant journal publications. 

2. Materials and Methods
Lines 118 - 119 - "high performance liquid chromatography (HPLC)" conditions and methodology use for the analysis should be described.
Lines 119 - 121 - "In the case of lignans, and phenolic acids in certain foods such as cereals, olives, and beans, data corresponding to HPLC after hydrolysis were used to calculate aglycone" - The methodology of hydrolysis should be included in the methods. Or even in the annexed section material.

The equations described in the text should be placed in the mathematical format of equation.

Lines 210, 259 - "phenols acids" - It should be written phenol acids. 

Line 346 - "been widely studied45." - Please, correct the sentence. It will be more rigorous if the authors cite more references and references which are as recent as possible.

The discussion cannot be based in references that are almost all reviews, research studies are needed for the discussion as well.

Round 2

Reviewer 2 Report

The improved version of the work presented is with better scientific quality.